

**$C_5$ glycolipids of heterocystous cyanobacteria track symbiont abundance in the diatom *Hemiaulus hauckii***
**across the tropical north Atlantic**
Nicole J. Bale[1], Tracy A. Villareal[2], Ellen C. Hopmans[1], Corina P.D. Brussaard[1], Marc Besseling[1], Denise
Dorhout[1], Jaap S. Sinninghe Damsté[1], Stefan Schouten[1]
[1]Department of Marine Microbiology and Biogeochemistry, NIOZ Royal Institute for Sea Research and Utrecht
University, Den Burg, 1790 AB, The Netherlands
[2]Marine Science Institute, The University of Texas at Austin, Port Aransas, TX, United States of America
*Correspondence to*: Nicole J. Bale (Nicole.bale@nioz.nl)
**Abstract.** Diatom-diazotroph associations (DDAs) include marine heterocystous cyanobacteria found as exo-
and endosymbionts in multiple diatom species. Heterocysts are the site of $N_2$ fixation and have a thickened cell
walls containing unique heterocyst glycolipids which maintain a low oxygen environment within the heterocyst.
The endosymbiotic cyanobacteria *Richelia intracellularis* found in species of the diatom genus *Hemiaulus* and
*Rhizosolenia* makes heterocyst glycolipids (HGs) containing pentose ($C_5$) moieties that are distinct from limnetic
cyanobacterial HGs with $C_6$ moieties. Here we applied a method for analysis of intact polar lipids (IPLs) to the
study of HGs in suspended particulate matter (SPM) and surface sediment from across the tropical North
Atlantic. The study focused on the Amazon plume region where DDAs are documented to form extensive
surface blooms in order to examine the utility of $C_5$ HGs as markers for DDAs as well as their transportation to
underlying sediments. $C_5$ HGs were detected in both marine SPM and surface sediments. We found a significant
correlation between the water column concentration of $C_5$ HGs and DDA symbiont counts. In particular, the
concentrations of both the $C_5$ HGs (1-(O-ribose)-3,27,29-triacontanetriol ($C_5$ $HG_{30}$ triol) and 1-(O-ribose)-
3,29,31-dotriacontanetriol ($C_5$ $HG_{32}$ triol)) in SPM exhibited a significant correlation with the number of
*Hemiaulus hauckii* symbionts. This result strengthens the idea that $C_5$ HGs can be applied as biomarkers for
marine endosymbiotic heterocystous cyanobacteria. The presence of the $C_5$ HGs in surface sediment provides
evidence that they are effectively transported to the sediment and hence have potential as biomarkers for studies
of the contribution of DDAs to the paleo-marine N-cycle.
**1 Introduction**
Cyanobacteria are cosmopolitan oxygenic photoautotrophs that play an important role in the global carbon and
nitrogen cycles. Marine cyanobacteria are the major fixers of dinitrogen ($N_2$) in modern tropical and subtropical
oligotrophic oceans (Karl et al., 1997; Lee et al., 2002). Because $N_2$ fixation is sensitive to oxygen, cyanobacteria
have evolved a range of different strategies in order to combine the incompatible processes of oxygenic
photosynthesis and $N_2$ fixation. One strategy, found only in filamentous cyanobacteria, is to fix $N_2$ in
differentiated cells known as heterocysts (Wolk, 1973; Rippka et al., 1979). Free-living heterocystous
cyanobacteria are rare in the open ocean (Staal et al., 2003); however, heterocystous taxa are abundant as both
exo- and endosymbionts in diatoms (Foster et al., 2011; Gómez et al., 2005; Luo et al., 2012; Villareal, 1991;




Villareal et al., 2011, 2012). These diatom-diazotroph associations (DDAs) can fully support the nitrogen (N)
requirements of both host and symbiont (Foster et al., 2011; Villareal, 1990) which explains the presence of
these symbioses in oligotrophic offshore environments such as the North Pacific gyre (Venrick, 1974). In the
western tropical north Atlantic Ocean, these symbiotic associations produce nearly 70% of total N demand in the
surface waters (Carpenter et al., 1999) as non-symbiotic diatom blooms deplete N in the Amazon River plume
and create N-poor conditions with residual P and Si (Subramaniam et al., 2008; Weber et al., 2017).

In all non-symbiotic cyanobacteria studied to date, the heterocyst cell walls contain heterocyst

glycolipids (HGs) (Abreu-Grobois et al., 1977; Bauersachs et al., 2009a, 2014; Gambacorta et al., 1995; Nichols
and Wood, 1968). These HGs comprise a hexose head group ($C_6$) glycosidically bound to long chain diols, triols,
or hydroxyketones (cf. Fig. 1) (Bauersachs et al., 2009b, 2011; Bryce et al., 1972; Gambacorta et al., 1998). In
contrast, the endosymbiotic heterocystous cyanobacterium *Richelia intracellularis* (found within the marine
diatoms *Hemiaulus hauckii* and *H. membranaceus;* (Villareal, 1991) contained HGs with a pentose sugar head
group ($C_5$) rather than a $C_6$ sugar (Fig. 1) (Schouten et al., 2013). The structural difference in the glycolipids of
marine endosymbotic heterocystous cyanobacteria compared to the free-living counterparts was hypothesized to
be an adaptation to the high intracellular $O_2$ concentrations within the host diatom (Schouten et al., 2013).

In the first study of the $C_5$ HGs in the natural environment, these compounds were found in suspended

particulate material (SPM) and surface sediment from the Amazon plume but not in lake sediments or river SPM
(Bale et al., 2015). However, HGs with a $C_5$ sugar moiety comprising a shorter $C_{26}$ carbon chain were tentatively
identified in a culture of freshwater cyanobacterium *Aphanizomenon ovalisporum* UAM 290 and in suspended
particulate matter from three freshwater environments in Spain (Wörmer et al., 2012). Thus, it remains to be
demonstrated whether distinctive $C_5$ HGs are unambiguously associated with DDAs in the marine environment.
In addition, the genera *Rhizosolenia*, *Guinardia* and *Hemiaulus* all contain species harboring heterocystous
cyanobacteria. DDA taxonomic relationships and host-symbiont specificity are only partially defined (Hilton et
al 2014, Foster and Zehr 2006, Janson et al 1999), suggesting additional clarification of how diverse HGs are
distributed within DDAs is required.

In this study, we applied a novel Ultra High Pressure Liquid Chromatography- High Resolution Mass

Spectrometry (UHPLC-HRMS) method to analyze the concentration of HG lipids in SPM from the oligotrophic
open Atlantic Ocean to the region affected by the Amazon River plume. We compared lipid concentrations with
the number of diazotrophic symbionts to examine the applicability of HGs to trace these organisms. Furthermore,
we also analyzed HG lipids in the surface sediment along the transect to examine the transport of these
compounds to the geological record and potential for use as a molecular tracer for DDA $N_2$ fixation.
**2 Methods**
**2.1 Cruise track and physiochemical parameters**
Sampling was carried out during a 4 week research cruise (64PE393) onboard the R/V *Pelagia* from 26th August
– 21st September 2014. The cruise followed a >5000 km transect and sampling occurred at 23 stations, starting
at Cape Verde and finishing at the island of Barbados (Fig. 2). The cruise track began close to the Cape Verde
EEZ boundary and proceeded approximately south-westerly across the Atlantic (Fig. 2). Aquarius sea-surface
salinity (SSS) satellite data (30 day composite, centered on 01-Sept-14) clearly indicated the influence of the





freshwater Amazon discharge in the region, i.e. surface salinity < 33 (Fig. 2). Discrete CTD measurements of
salinity (contour lines Fig. 3a, Table 1) generally agreed with the satellite data as to the geographical spread of
the Amazon River plume. However, the region was highly dynamic with the plume location shifting hundreds of
km over the course of the cruise as noted in the sequential 7-day Aquarius SSS composites (Fig. S1 -
Supplemental material).
Temperature and salinity were measured using a Sea-Bird SBE911+ conductivity–temperature–depth
(CTD) system equipped with a 24 × 12 L Niskin bottles rosette sampler. Fluorescence was measured with a
Chelsea Aquatracka MKIII fluorometer. Chlorophyll fluorescence was not calibrated against discrete chlorophyll
and is reported as relative fluorescence units (RFU). Seawater samples for dissolved inorganic nutrient analysis
were taken from the Niskin bottles in 60 ml high-density polyethylene syringes with a three way valve and
filtered over Acrodisc PF syringe filters (0.8/0.2 µm Supor Membrane, PALL Corporation ) into pre-rinsed 5 mL
polyethylene vial. Dissolved orthophosphate ($PO_4$) and nitrogen ($NO_3$, $NO_2$ and $NH_4$) were stored in dark at 4°C
until analysis onboard (within 18 h) using a QuAAtro autoanalyzer (Grasshoff, 1983; Murphy and Riley, 1962).
Samples for dissolved reactive silicate (Si) analysis (Strickland and Parsons, 1968) were stored dark at 4°C until
analysis using the same system as above upon return to NIOZ. The detection limits were calculated as: $PO_4$
0.004 µmol L$^{-1}$, $NH_4$ 0.030 µmol L$^{-1}$, $NO_3$+$NO_2$ 0.005 µmol L$^{-1}$ and $NO_2$ 0.002 µmol L$^{-1}$.
**2.2 Phytoplankton pigment composition and enumeration of diazotrophs**
Samples for diazotroph enumeration were collected in polycarbonate bottles of which 500-1170 ml was filtered
under gentle vacuum (< 5 psi) through a 10 µm pore-size polycarbonate filter (47 mm diameter). Filters were
placed onto 75 X 50 mm glass slides (Corning 2947) and 2-3 drops of non-fluorescent immersion oil (Cargille
type DF) placed on the slide. A glass cover slip (45 x 50 mm; Fisherbrand 12-545-14) was placed on the filter
sample and allowed to sit while the immersion oil cleared the filter. The sample was subsequently viewed under
transmitted light and epi-fluorescence illumination light filter (530-560 nm excitation, 572-648 nm emission;
Olympus BX51) for counting/identifying trichomes and host cells as well as photomicrography (Olympus
DP70).
For phytoplankton pigment analysis, seawater was filtered through 0.7 µm glass fiber GF/F filters (Pall
Corporation, Washington). The filters were extracted in 4 mL 100% methanol buffered with 0.5 mol L$^{-1}$
ammonium acetate, homogenized for 15 s, and analyzed by high performance liquid chromatography (HPLC).
The relative abundances of the different taxonomic groups were determined using CHEMTAX (Mackey et al.,
1996; Riegman and Kraay, 2001).
**2.3 SPM and surface sediment collection**
Three McLane *in situ* pumps (McLane Laboratories Inc., Falmouth) were used to collect suspended particulate
matter (SPM) from the water column for lipid analysis. They were generally deployed at three depths: the
surface (3 - 5 m), the bottom wind mixed layer (BWML) and the deep chlorophyll-*a* maximum (DCM), with
some additional sampling at 200 m (Table 1). They pumped between 90 and 380 L with a cut-off at a pre-
programmed pressure threshold and the SPM was collected on pre-ashed 0.7 µm, 142 mm, GF/F filters (Pall
Corporation, Washington) and immediately frozen at −80°C. At Sta., as part of a different study (Besseling et al





in prep), 12 additional sampling points were carried out to produce a high resolution depth profile (Table 2)
where the SPM was collected on pre-ashed 0.3 µm GF75 filters (Avantec, Japan).
Sediment was collected at each station in 10 cm diameter, 60 cm length multicores. For sediment
sampling site, triplicate cores were collected, always from a single multicore deployment (with a maximum of 60
cm between core centers). The cores were sliced into 1 cm slices using a hydraulic slicer and each slice was
stored separately in a geochemical bag and immediately frozen at –80°C. For this study we analyzed the 0–1 cm
(surface sediment) slice. For TOC analysis, sediment was freeze dried and analysis was carried out using a Flash
2000 series Elemental Analyzer (Thermo Scientific) equipped with a TCD detector.

**2.4 Lipid extraction**

The extraction of lipids from freeze dried filtered seawater or sediment samples was carried out using a modified
Bligh-Dyer extraction (Bale et al., 2013). The samples were extracted in an ultrasonic bath for 10 min with 5 –
20 ml of single-phase solvent mixture of methanol (MeOH): dichloromethane (DCM): phosphate buffer (2:1:0.8,
v:v:v). After centrifugation ($1000 \times g$ for 5 min, room temperature, Froilabo Firlabo SW12 with swing out rotor)
to separate the solvent extract and residue, the solvent mixture was collected in a separate flask. This was
repeated three times before DCM and phosphate buffer were added to the single-phase extract to induce phase
separation, producing a new ratio of MeOH:DCM:phosphate buffer (1:1:0.9 v:v:v). After centrifugation ($1000 \times$
$g$ for 5 min), the DCM phase was collected in a glass round-bottom flask and the remaining MeOH:phosphate
buffer phase was washed two additional times with DCM. Rotary evaporation was used to reduce the combined
DCM phase before it was evaporated to dryness under a stream of $N_2$.

**2.5 Analysis of intact polar lipids**

Whereas previous studies of heterocyst glycolipids have applied high performance liquid chromatography
multiple reaction monitoring (MRM) mass spectrometry (HPLC–MS[2]) method (e.g., Bale et al. (2015)), in this
study we used an Ultra High Pressure Liquid Chromatography-High Resolution Mass Spectrometry (UHPLC-
HRMS) method, designed for the analysis of a wide range of intact polar lipids (Moore et al., 2013). The
UHPLC-HRMS method was adapted by replacement of hexane with heptane as the non-polar solvent in the
eluent, to reduce the toxic nature of hexane relative to heptane in terms of a work place health hazard (Buddrick
et al., 2013; Carelli et al., 2007; Daughtrey et al., 1999). Our UHPLC-HRMS method was as follows: we used an
Ultimate 3000 RS UHPLC, equipped with thermostatted auto-injector and column oven, coupled to a Q Exactive
Orbitrap MS with Ion Max source with heated electrospray ionization (HESI) probe (Thermo Fisher Scientific,
Waltham, MA). Separation was achieved on an Acquity UPLC BEH HILIC column (150 x 2.0 mm, 2.1 µm
particles, pore size 12 nm; Waters, Milford, MA) maintained at 30 °C. Elution was achieved with (A) heptane-
propanol-formic acid-14.8 mol L[-1] aqueous $NH_3$ (79:20:0.12:0.04, v/v/v/v) and (B) propanol water-formic acid-
14.8 mol L[-1] aqueous $NH_3$ (88:10:0.12:0.04, v/v/v/v) starting at 100% A, followed by a linear increase to 30% B
at 20 min, followed by a 15 min hold, and a further increase to 60% B at 50 min. Flow rate was 0.2 ml min[-1],
total run time was 70 min, followed by a 20 min re-equilibration period. Positive ion ESI settings were: capillary
temperature, 275°C; sheath gas ($N_2$) pressure, 35 arbitrary units (AU); auxiliary gas (N2) pressure, 10 AU; spray
voltage, 4.0 kV; probe heater temperature, 275°C; S-lens 50 V. Target lipids were analyzed with a mass range of
$m/z$ 350–2000 (resolution 70,000 ppm), followed by data-dependent tandem MS[2] (resolution 17,500 ppm), in





which the ten most abundant masses in the mass spectrum were fragmented successively (normalized collision
energy, 35; isolation width, 1.0 $m/z$). The Q Exactive was calibrated within a mass accuracy range of 1 ppm
using the Thermo Scientific Pierce LTQ Velos ESI Positive Ion Calibration Solution. During analysis dynamic
exclusion was used to temporarily exclude masses (for 6 s) in order to allow selection of less abundant ions for
$MS^2$. In addition, an inclusion list (within 3 ppm) was used, containing all known HGs, in order to obtain
confirmatory fragment spectra.
Before analysis, the extracts were re-dissolved in a mixture of heptane, isopropanol and water (72:27:1,
v:v:v) which contained two internal standards (IS), a platelet-activating factor (PAF) standard (1-O-hexadecyl-2-
acetyl-sn-glycero-3-phosphocholine, 5 ng on column) and a short-chain glycolipid standard, n-dodecyl-β-D-
glucopyranoside (≥98% Sigma-Aldrich, 20 ng on column; cf. Bale et al. (2017)). The samples were then filtered
through 0.45 μm mesh True Regenerated Cellulose syringe filters (4 mm diameter; Grace Alltech). The injection
volume was each sample was 10 μl. For quantification the relative response factor (RRF) between the n-dodecyl-
β-D-glucopyranoside IS and an isolated $C_6$ HG (1-(O-hexose)-3,25-hexacosanediol (Bale et al., 2017) was
determined to be 6.63.
The 12 samples collected at Sta. 10 (0.3 μm GF75 filters, Table 2) were analyzed on the same UHPLC-
HRMS system, but with hexane instead of heptane in the mobile phase. Also, the n-dodecyl-β-D-
glucopyranoside IS was not added, so quantification was based the PAF IS and correcting for the RRF between
the n-dodecyl-β-D-glucopyranoside IS and the PAF IS.
**2.6 Statistical analysis**
T-tests and Pearson correlations were determined using Sigmaplot software (version 13.0).
**3. Results**
**3.1 Physicochemical conditions and phytoplankton assemblage**
Stations 1-6, 12 and 22 correspond to oceanic stations (SSS > 35, following the convention of Subramaniam et
al. (2008)), with Sta. 7-11 in the intermediate salinity range (30 – 35). Originally termed mesohaline
(Subramaniam et al., 2008), we use 'intermediate salinity' to avoid confusion with the older use of mesohaline in
coastal systems to refer to 5-18 waters (Elliott and McLusky, 2002). Only Sta. 11, with a SSS of 29.2, was in the
low salinity range defined by Subramaniam et al. (2008). Temperature was uniformly high across the cruise track
(>27ºC in the euphotic zone) with the 25° C isotherm deepening along the cruise track (Fig. 3a). Stations 1-6
exhibited typical tropical open-ocean conditions with an average 36.3 ± 0.2 and depleted surface (3 – 5 m)
inorganic nutrient concentrations (on average 0.01 ± 0.01 μmol L$^{-1}$ PO$_4$, 0.02 ± 0.20 μmol L$^{-1}$ NO$_{3+}$NO$_2$ and 0.86
± 0.09 μmol L$^{-1}$ Si (Fig. 3c,d). Surface NO$_{3+}$NO$_2$ and PO$_4$ concentration remained low at Sta. 7 – 11 (Fig. 3c),
although PO$_4$ increased slightly to 0.07 μmol L$^{-1}$. Si concentrations increased > 10-fold at Sta. 7-11 and was on
average of 12.1 ± 4.4 μmol L$^{-1}$ (Fig. 3d). Sta. 12 was shallow and close to the coast but was just north of the
point of plume retroflection (cf. Fig. 2), as evidenced by an increased SSS (35.4) and relatively lower Si
concentration (3.14 μmol L$^{-1}$). The deep chlorophyll (Chl) maximum (DCM; cf. maxima in chl fluorescence
(Fig. 3b) was associated with the nutricline (cf. Fig. 3c) over most of the transect, with the highest DCM
fluorescence at the oceanic stations Sta. 1 and 2. There was a secondary surface Chl fluorescence maximum at





Sta. 11 which was the most nearshore, lowest salinity station. Just north of the plume, Station 12 displayed a
more mixed water column profile with uniformly elevated Chl fluorescence to ~75 m (Fig. 3b).
From the coastal shelf of French Guiana (Sta. 11 and 12), the cruise progressed in a northerly direction
towards the Caribbean. The Amazon River influence was again evident after Sta. 13, but decreased with
distance, with SSS ranging from 32.8 at Sta. 13 to a maximum of 35.6 at Sta. 22. Surface $NO_{3+}NO_2$ remained
low through Sta. 13 – 23 (on average $0.01 \pm 0.00$ µmol L$^{-1}$), while $PO_4$ was variable but generally decreased to
open ocean levels (from 0.01 µmol L$^{-1}$ at Sta. 13 to below the limit of detection at Sta. 23). Si dropped from 10.4
µmol L$^{-1}$ at Sta. 13 to 3.94 µmol L$^{-1}$ at Sta. 23.
The phytoplankton pigment composition analysis at Sta. 1-6 was dominated by the cyanobacteria
*Prochlorococcus* which made up around 50% of total Chl a in the surface waters (Table S1). At Sta. 7-10 and
18-23, the phycoerythrin-containing cyanobacteria (e.g. *Synechococcus*) dominated the phytoplankton
community. In general, at Sta. 7-10 the share of Chrysophytes and Prymnesiophycea pigments was relatively
larger. The share of Chrysophyceae was particularly large at the DCM, even dominating the phytoplankton
community biomass at Sta. 15-23 (Table S1). Diatoms (Bacillariophyceae) contributed substantially in the
surface waters of Sta. 8, up to 21% of total Chl a.

**3.2 Diazotroph enumeration**

The diazotroph cyanobacteria were divided into 5 categories: three of them are symbionts, i.e. with the diatoms
*Rhizosolenia cf. clevei*, *Hemiaulus hauckii,* and *Guinardia cylindrus* DDAs, and two are non-symbionts, i.e.
*Trichodesmium* colonies (>10 trichomes organized into a coherent structure), and free *Trichodesmium* trichomes
(Fig. 4a-e, Table S3). Total DDA abundance was low (0-21 combined DDA *Richelia* trichomes L$^{-1}$) at Sta. 1-6.
*Hemiaulus* DDA abundance was greatest at Sta. 8 (ca. 4.0 x10$^3$ trichomes L$^{-1}$) with a secondary maximum at Sta.
17 (0.8 x10$^3$ trichomes L$^{-1}$), both in the surface (<5 m) waters. *Rhizosolenia* DDA abundance was lower than
*Hemiaulus* DDA abundance at Sta. 7 (*Rhizosolenia* DDA ca. 60 trichomes L$^{-1}$) and at Sta. 15 and 16
(*Rhizosolenia* DDA, ca. 80 trichomes L$^{-1}$). *Rhizosolenia* DDAs were not observed below 31.6 salinity (Fig. 4a).
*Hemiaulus* DDA were observed down to 27.1-27.6 salinity at ~80-100 trichomes L$^{-1}$ (Fig, 4b). Free
*Trichodesmium* trichomes were broadly distributed (Fig. 4d). Free *Trichodesmium* trichomes often occurred
across a wide depth range, down to 75 m at Sta. 17. *Trichodesmium* colonies were seen sporadically and with
distributions dominated by two sampling points (Sta. 6, 32 m and Sta. 21, 60 m) where colony abundance > 25
colonies L$^{-1}$. A single observation of colonies at depth under the low salinity plume generated contour lines
suggesting a generalized presence at depth. However, removal of this observation (Sta. 14, 61 m) removed this
trend and resulted in distinct separation of the colony distributions, i.e. two areas of increased biomass associated
with salinity gradients at the edge of the river plume.

**3.3 Heterocyst glycolipids in suspended particulate matter**

We analyzed heterocyst glycolipids (HGs) in SPM from along the cruise transect collected at the surface,
bottom wind mixed layer (BWML) and the DCM. Two $C_5$ HGs were detected in the SPM, i.e. 1-(O-ribose)-
3,27,29-triacontanetriol and 1-(O-ribose)-3,29,31-dotriacontanetriol ($C_5$ HG$_{30}$ and $C_5$ HG$_{32}$ triol respectively,
Fig. 1). $C_5$ HG$_{32}$ triol represented on average 98 % $\pm$ 4 of the combined concentration of the two HGs. Previous
studies of $C_5$ HGs have identified 1-(O-ribose)-3,29-triacontanediol ($C_5$ HG$_{30}$ diol, Fig. 1) in both cultures and





environmental samples (Bale et al., 2015; Schouten et al., 2013), but these were not seen in the SPM or surface
sediment analyzed in this study.

The concentrations of the two $C_5$ HGs were highest in the surface waters of Sta. 8 and showed a second local
maxima at Sta. 16 (Table 1 and Fig. 4f). The surface concentration of the dominant HG, i.e. $C_5$ $HG_{30}$ triol, ranged
between 0 and 4800 pg $L^{-1}$. The range in concentration was 50-fold lower at the DCM (0-200 pg $L^{-1}$, Table S2).
The three samples from 200 m depth showed lowest concentrations, ranging between 20.6 and 127 pg $L^{-1}$.
Overall, the $C_5$ $HG_{30}$ triol was consistently present in the higher concentration of the two (Fig. 4f). The minor
HG, i.e. $C_5$ $HG_{32}$ triol, ranged between 0 – 10% of their combined concentration at the surface and BWML, was
between 0 – 5% at the DCM and 0 – 17% at 200 m (cf. Fig. 4f contour lines and Table S2).
HGs with a $C_6$ sugar head group were not detected in any SPM samples with the exception of one
sample, taken at Sta. 20a from the DCM (65 m). 1-(O-hexose)-3,25-hexacosanediol ($C_6$ $HG_{26}$ diol, Fig. 1) and 1-
(O-hexose)-3-keto-25-hexacosanol ($C_6$ $HG_{26}$ keto-ol) were confidently identified from their $[M+H]^+$ accurate
mass ($m/z$ 577.4674 and 575.4517 respectively) and their fragmentation patterns, which followed published
reports (Bauersachs et al., 2009b). $C_6$ $HG_{26}$ diol and $C_6$ $HG_{26}$ keto-ol were present at concentrations of 0.3 and
0.4 ng $L^{-1}$ respectively (data not shown), both ~10 times higher than the concentration of the $C_5$ $HG_{30}$ triol in this
sample (Table 1).
At Sta. 10, besides SPM samples collected on 0.7 µm GF/Fs, SPM samples were also collected at
depths down to 3000 m using 0.3 µm GF75 filters (Table 2). As with the 0.7 µm SPM samples at station 10, $C_5$
$HG_{30}$ triol was consistently present in higher concentration than $C_5$ $HG_{32}$ triol (which represented on average
only 1.4 % ± 0.7 of their combined concentration). The concentrations and depth trends (to 200 m) of the two $C_5$
HGs did not differ between the 0.3 µm and 0.7 µm filter SPM samples (Fig. 5). For both the 0.3 µm samples and
the 0.7 µm samples, the combined concentration of $C_5$ $HG_{30}$ triol and $C_5$ $HG_{32}$ triol was highest at 200 m, 108 pg
$L^{-1}$. In the 0.3 µm samples, both concentrations decreased below 200 m, although both $C_5$ HGs remained
detectable at 3000 m depth.

### 3.4 Heterocyst glycolipids and bulk properties in surface sediment

As with the SPM, $C_5$ $HG_{30}$ triol and $C_5$ $HG_{32}$ triol were detected in the surface sediment of seventeen stations
(Table 1). $C_5$ $HG_{30}$ triol was also here consistently present in the higher concentration of the two ($C_5$ $HG_{32}$ triol
represented on average 9.4 % ± 3.0 of their combined concentration). The $C_5$ $HG_{30}$ diol was not detected in any
surface sediment, alike the SPM samples. HGs with a $C_6$ sugar head group were also not detected in any surface
sediment. In the sediment underlying the high-salinity open ocean stations (1, 3, 5) the combined concentration
of the two $C_5$ HGs was low (2.0 – 3.7 ng $g^{-1}$, Table 1). It was high at Sta. 7 and 8 (10.6 and 16.3 ng $g^{-1}$), while
Sta. 9 – 17 contained mid-range concentrations (5.2 – 14.8 ng $g^{-1}$), with the exception of the two coastal-shelf
stations (11 and 12) where the concentration was at its lowest (0.2 and 0.3 ng $g^{-1}$). At the final 4 stations (20a,
21a, 22 and 23) the combined concentration returned to high levels (11.2 – 19.0 ng $g^{-1}$). For context, the TOC
was relatively stable between Sta. 1 and 10 (av. 0.6 ± 0.1 %, n=7) then low at Sta. 11 and 12 (av. 0.2 ± 0.1 %).
Sta. 13 exhibited the highest TOC of all the stations (1.2 ± 0.0 %), and TOC decreased steadily at all stations
thereafter, and was 0.6 ± 0.0 % at Sta. 23.



## 4. Discussion

### 4.1 Heterocyst glycolipids and DDAs in the water column

The Amazon plume has been extensively documented to support high numbers of the diatom-diazotroph associations (DDA) such as *Hemiaulus hauckii- Richelia intracellularis* and *Rhizosolenia clevei-Richelia intracellularis* (Carpenter et al., 1999; Foster et al., 2007; Goes et al., 2014; Subramaniam et al., 2008; Weber et al., 2017). Our study took place outside the high Amazon flow period and the Chl concentrations and DDA counts encountered on this cruise did not reach the values seen in 'bloom conditions' described during previous studies in the region (Carpenter et al., 1999; Subramaniam et al., 2008). However, the DDA counts in certain stations were up to 3 orders of magnitude higher than surrounding waters and comparable to the open ocean DDA blooms seen in the North Pacific gyre (Villareal et al., 2011, 2012). These strong gradients permitted to investigate relationships between DDA and HG distributions.

The concentration of the $C_5$ HGs were correlated with the cell counts of different diazotrophs. The concentrations of both the $C_5$ HGs (1-(O-ribose)-3,29,31-dotriacontanetriol ($C_5$ $HG_{30}$ triol, Fig. 1) and 1-(O-ribose)-3,27,29-triacontanetriol ($C_5$ $HG_{32}$ triol) exhibited the most significant positive Pearson correlation with the number of *Hemiaulus* symbionts ($p \leq 0.001$, $r = 0.79$ and $0.78$ respectively, n=54). While $C_5$ heterocyst glycolipids (HGs) have been found in cultures of DDAs (Bale et al., 2015; Schouten et al., 2013), our study of the tropical north Atlantic provides to our knowledge for the first time, environmental evidence that $C_5$ HGs track the abundance and distribution of DDAs.

Interestingly, there was no significant correlation found between the number of *Rhizosolenia* symbionts and the concentration of the $C_5$ HGs ($C_5$ $HG_{30}$ triol: $p = 0.07$, $r = 0.23$ and $C_5$ $HG_{32}$ triol: $p = 0.14$, $r = 0.19$), except when the surface and BWML of Sta. 8 were excluded from the analysis ($C_5$ $HG_{30}$ triol: $p \leq 0.001$, $r = 0.88$; and $C_5$ $HG_{32}$ triol: $p \leq 0.001$, $r = 0.83$). This difference may in part be due to the lower number of *Rhizosolenia*/*Guinardia* symbionts relative to *Hemiaulus* symbionts (on average *Rhizosolenia* symbionts in this study represented $24 \pm 34$ % of the sum of *Rhizosolenia* and *Hemiaulus* symbionts), similar to previous findings that *Hemiaulus* dominated over *Rhizosolenia* in the Amazon plume (Foster et al., 2007) and Caribbean region (Villareal, 1994). Furthermore, culture studies have shown that *Rhizosolenia* symbionts contain only trace amounts of $C_5$ $HG_{30}$ triol, (Bale et al., 2015)**,** whereas this is a dominant HG in *Hemiaulus* symbionts (Schouten et al., 2013). In this study, *Rhizosolenia* DDAs were not observed below salinities of 31.5, while *Hemiaulus* DDAs were observed at salinities of 27.5 (Fig. 6). The plume is highly dynamic and it is unclear whether this is a significant niche separation between the taxa or simply mixing and loss of the less abundant *Rhizosolenia* DDA at the water volumes being counted.

Unfortunately, a unique biomarker for *Rhizosolenia* and *Guinardia* symbionts has not been identified to date (Bale et al., 2015; Schouten et al., 2013). There was also a significant correlation between $C_5$ $HG_{32}$ triol, (but not $C_5$ $HG_{30}$ triol) and the counts of *Guinardia cylindrus (*formerly *Rhizosolenia cylindrus)* ($p \leq 0.03$, $r = 0.49$, n=21). This was the only species for which there was a correlation with $C_5$ $HG_{32}$ triol but not $C_5$ $HG_{30}$ triol**.** This DDA has not been cultured and nothing is known about the heterocyst lipid composition of this species. These results suggest $C_5$ $HG_{30}$ triol may be synthesized by this species.

At approximately half of the sampling points, glycolipids could be detected in SPM where no DDAs were observed by microscopy. These sampling points generally contained low combined concentration of the two $C_5$ HG lipids ($0 - 62.9$ pg $L^{-1}$, n=22), compared to the sampling points where DDAs were detected (18 -





5300 pg L$^{-1}$, n=32). These two groups were significantly different from each other (as determined by t-test, $p =$
<0.001). This difference may be result of the difference in total sampling volumes between the two methods.
Microscopic examinations were carried out using 0.5 - 1.2 L per sample. Although for lipid analysis 90 - 400 L
were filtered, the individual analyses on the UHPLC-HRMS system each represented between 0.5 and 5 L of
seawater. However, the far greater initial sample volume leads to a higher probability that the lipid samples
would contain symbiont chains than the microscopy samples. In addition, microscopic examinations may have
missed free heterocysts and heterocysts that were incorporated into unrecognizable masses in aggregates whereas
UHPLC-HRMS may have still detected the associated HGs. Indeed, copepod grazing in the plume (Conroy et
al., 2016) will repackage *Richelia* trichomes, and little is known of the effects of gut passage on heterocyst and
HG integrity. It should also be noted that because sampling for diazotroph enumeration and for lipid analysis
occurred via different methods, , there was a time offset of ≤ 5 h and a depth offset of ≤ 20 m between the two
sampling events representing the same water column phenomena (surface, BWML and DCM)
Unexpectedly, a significant correlation was also found for $C_5$ $HG_{30}$ triol and $C_5$ $HG_{32}$ triol and the
number of *Trichodesmium* colonies ($p ≤ 0.001$, $r = 0.68$ and 0.67, n=54), and for $C_5$ $HG_{30}$ triol and the number of
*Trichodesmium* filaments ($p ≤ 0.05$, $r = 0.30$, n=54). These correlations could be coincidental as $C_5$ HG
producing organisms have not been described in association with *Trichodesmium* nor would *Trichodesmium* be
expected to produce HGs itself as it does not use heterocysts to fix nitrogen. A recent study in the North Pacific
Subtropical Gyre found that *Trichodesmium* colonies were harboring an endobiontic heterocystous cyanobacteria
of the genus *Calothrix* (Momper et al., 2015). However, analyses of the HG content of both freshwater and
marine *Calothrix* cultures have to date only revealed the presence of $C_6$ HGs, not $C_5$ HGs (Bauersachs et al.,
2009a; Schouten et al., 2013; Wörmer et al., 2012). Furthermore, no heterocystous cyanobacteria were observed
in *Trichodesmium* from the Caribbean (Borstad, 1978) or southwest Sargasso Sea (Siddiqui et al., 1992).
*Trichodesmium* is reported to have a physiological differentiated cell (diazocyte) that permits $N_2$-fixation in an
oxygenated colony or trichome, and which lacks the thickened cell envelope of heterocysts where HGs are
localized (Sandh et al., 2012).
While elevated HGs were statistically more associated with the DDA blooms than either free or
colonial *Trichodesmium*, there was frequently a co-occurrence of *Trichodesmium* with the DDA taxa (Fig. 4)
which could also contribute to the unexpected correlation. The *Trichodesmium* distribution appears to contrast
with the findings of Foster et al. (2007), Goes et al. (2014) and Subramaniam et al. (2008), who all concluded
that changing nutrient availability as reflected in the salinity gradient along the Amazon River plume led to
zonation of the diazotroph community. However, their data were examining more pronounced DDA cell
abundance concentrations under much higher Amazon plume flow conditions. The broader features of our
observations, i.e. a low salinity region with higher nutrient concentrations and few diazotrophs transitioning to
strong diazotroph gradients in the salinity gradient to oceanic conditions, are in concordance with their
observations.
Two $C_6$ HGs, generally associated with free-living heterocyst forming cyanobacteria from freshwater or
brackish environments (Bale et al., 2015, 2016, Bauersachs et al., 2009b, 2010, 2011; Bühring et al., 2014;
Wörmer et al., 2012) were identified only in the DCM of Sta. 20a ($C_6$ $HG_{26}$ diol and $C_6$ $HG_{26}$ keto-ol). Whereas
in this study the two $C_6$ HGs were found at a similar concentration to each other, previous studies have reported
that $C_6$ $HG_{26}$ keto-ol was detected a minor component relative to the more abundant $C_6$ $HG_{26}$ diol (Bale et al.,





2015, 2016, Bauersachs et al., 2009a, 2009b, 2011; Schouten et al., 2013; Wörmer et al., 2012). An earlier study
executed nearer to the mouth of the Amazon river detected trace levels of $C_6$ $HG_{26}$ diol (but not $C_6$ $HG_{26}$ keto-ol)
in surface sediments (Bale et al., 2015). In contrast, both $C_6$ $HG_{26}$ diol and $C_6$ $HG_{26}$ keto-ol were recorded in
freshwater Amazon River water and floodplain lake sediment.

There are reports of cyanobacterial species in cohabitation with other vegetal such as the floating

macroalgae *Sargassum* (Carpenter, 1972; Hanson, 1977; Phlips et al., 1986) and *Trichodesmium* (Momper et al.,
2015). While the HG content of the cyanobacteria in these co-habitations has not been investigated, these
cyanobacteria are in the same families as known $C_6$ HG producers (Bauersachs et al., 2009a; Schouten et al.,
2013; Wörmer et al., 2012). *Trichodesium* was not detected by microscopy at this sampling point, however as
stated above, there is an apparent difference the limit of detection between counting by microscopy and lipid
analysis by UHPLC-HRMS. Floating 'fields' of *Sargassum* were regularly encountered during the research
cruise, with the maximum observations occurring around Sta. 16 (pers. obs.). Further work on the HG
composition of the cyanobacteria found in these cohabitations would be necessary to draw conclusions as to
whether they contributed to the source of the two $C_6$ HGs detected at this sampling point.

**4.2 $C_5$ Heterocyst glycolipids below the DCM**


While the concentration of the $C_5$ HGs was generally highest within the mixed layer (ML, cf. Fig. 4f), Sta. 10
exhibited an increase in $C_5$ HG concentration with depth with $C_5$ HGs in both the 0.3 μm and 0.7 μm samples
increasing with depth to a maximum at 200 m (Fig. 5). The two size fraction profiles were carried out
approximately 12 hours apart and suggests that the HG maxima at 200 m was a feature for at least this period of
time. Sta. 9 was the only other station where the $C_5$ HG concentration (0.7 μm) at the DCM was higher than in
the ML (cf. Table S1). Foster et al. (2007) reported that DDAs are high in the ML but can increase below the ML
down to at least 100 m. Sediment trap studies in the North Pacific and tropical North Atlantic ocean have found
significant contributions by DDAs to the vertically exported particulate organic carbon (Karl et al., 2012;
Scharek et al., 1999; Subramaniam et al., 2008). While our study did not utilize sediment traps to collect sinking
particles, a proportion of the matter collected by in situ filtration is probably sinking rather than suspended
(Abramson et al., 2010). $C_5$ HGs have been found in surface sediment at depths up to 3000 m underlying our
water column sampling points (this study and Bale et al. (2015)), supporting the hypothesis that DDAs are
effectively transported in this environment from the water column to the sediment. These sinking particles could
be due to bloom-termination and aggregation or sinking of zooplankton fecal pellets.

**4.3 $C_5$ Heterocyst glycolipids in surface sediment**


As was found in a previous study concentrating on a smaller area close to the mouth of the Amazon (Bale et al.,
2015), the presence of a similar distribution of $C_5$ HGs in SPM and surface sediment indicates that HG producers
sink, probably enhanced by the mineral ballast as well as matrix protection provided by the association with
diatom silica skeletons. The total $C_5$ HG concentration in surface sediments was more spatially homogenous than
the distribution in the SPM (Table 1). Other than the two stations very close to the coast (where currents were
high and % TOC was at its lowest), the HGs were detected in comparably high levels from Sta. 7 onwards. This
reflects the wide spatial range of the HG-producers through an 'integrated' multi-decadal record of their
deposition. Each year between June and January, the Amazon plume is retroflected offshore, across the Atlantic



towards Africa due to the actions of the North Brazil Current and the North Equatorial Countercurrent, which may account for the presence of the $C_5$ HGs in the surface sediments of Sta. 1 - 10. The rest of the year the Amazon water flows northwestward towards the Caribbean Sea as the countercurrent and the retroflection weaken or vanish (Muller-Karger et al., 1988), in turn accounting for the $C_5$ HGs in the surface sediments of Sta. 13 - 23.

**5. Conclusions**

$C_5$ HGs were detected in the water column of the tropical North Atlantic and their concentrations correlated strongly with DDAs. Furthermore, the HGs tracked the movement of the DDAs to the surface sediments in areas known to be impacted by high seasonal DDA input (under the Amazon plume) whereas the HG concentration in sediment farther away from plume was low. We conclude that $C_5$ HGs provide a robust, reliable method for detecting DDAs in the marine environment. The apparent stability and specificity of $C_5$ HGs mean that they have high potential for use in future work examining the presence and N-cycling role of DDAs in the past.

**Acknowledgments**

We thank the captains and crew of the R/V *Pelagia* for their support during the cruise. We thank Yvo Witte and Sander Asjes for technical support onboard and Sharyn Ossebaar for nutrient sample collection and analysis. We thank colleagues from the MMB lab (NIOZ) for assistance with sample collection and processing. We also thank Steven de Vries for $C_6$ HG data analysis. The work of N. Bale is supported by the Netherlands Organisation for Scientific Research (NWO) through grant 822.01.017 to S. Schouten. SS was funded by the European Research Council (ERC) under the European Union's Seventh Framework Program (FP7/2007-2013) ERC grant agreement [339206]. S.S. and J.S.S.D. receive financial support from the Netherlands Earth System Science Centre (NESSC).

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





**Figure legends**
**Figure 1.** Structures of the heterocyst glycolipids detected in this study $C_6$ glycolipids: 1-(O-hexose)-3,25-
hexacosanediol ($C_6$ $HG_{26}$ diol), 1-(O-hexose)-3-keto-25-hexacosanol ($C_6$ $HG_{26}$ keto-ol). $C_5$ glycolipids: 1-(O-
ribose)-3,29-triacontanediol ($C_5$ $HG_{30}$ diol), 1-(O-ribose)-3,27,29-triacontanetriol ($C_5$ $HG_{30}$ triol), 1-(O-ribose)-
3,29,32-dotriacontanetriol ($C_5$ $HG_{32}$ triol). Grey box indicates glycolipids associated with DDAs

**Figure 2.** Map of tropical North Atlantic showing the study site. Location of the stations indicated. Aquarius sea-
surface salinity (SSS) satellite data from ERDAPP (30 day composite, centered on 01-Sept-14,
https://coastwatch.pfeg.noaa.gov/erddap/index.html ).

**Figure 3.** Water column characteristics along the cruise track. Color scales show a) temperature, b) chlorophyll
fluorescence (from fluorometer on CTD), c) $PO_4$ (color scale) and d) Si. Contour lines show salinity (a, b, d) and
$NO_3 + NO_2$ (c). Station numbers noted above plots, distance along transect from the Cape Verde Islands below.

**Figure 4.** Diazotroph abundance along the cruise track. Color scales show a) *Rhizosolenia* symbionts (trichomes
$L^{-1}$), b) *Hemiaulus* symbionts (trichomes $L^{-1}$), c) *Guinardia* symbionts (trichomes $L^{-1}$), d) *Trichodesmium* (free
trichomes $L^{-1}$) and e) *Trichodesmium* (colonies $L^{-1}$) while contour lines show salinity (a – e). f) Color scale
shows concentration of $C_5$ $HG_{30}$ triol (pg $L^{-1}$) while contour lines show $C_5$ $HG_{32}$ triol % (of $C_5$ total sum). Station
numbers above plots, distance along transect from the Cape Verde Islands below. Dots in Fig. 4a-c indicate
sampling depth for the salinity contours. Fig. 4d-e indicate sampling depth for HG lipids (Fig. 4f.). See
comments in text regarding Trichodesmium colony contouring artifacts.

**Figure 5.** Station 10, down column profile of $C_5$ HG sum ($C_5$ $HG_{30}$ triol + $C_5$ $HG_{32}$ triol, pg $L^{-1}$) from 0.7 μm
GF/F filters (grey broken line) and 0.3 μm GF75 filters (solid black line).

**Figure 6.** Plots of cell numbers and HG concentrations with a color scale showing salinity. a) *Rhizosolenia*
symbionts (trichomes $L^{-1}$), b) *Hemiaulus* symbionts (trichomes $L^{-1}$), c) $C_5$ $HG_{30}$ triol (pg $L^{-1}$), d) $C_5$ $HG_{32}$ triol
(pg $L^{-1}$) . Station numbers on x axis.







Table 1. Glycolipid concentrations from sea surface (3 – 5m) and surface sediment for all stations. For
concentrations at BWML and DCM see Table S3. † = No sediment collected, ns = not sampled.

| Station | Lat | Long | Date | Water depth (m) | Salinity | Sea surface | | Surface sediment | | |
|---|---|---|---|---|---|---|---|---|---|---|
| | | | | | | $C_5 HG_{30}$ triol (pg L$^{-1}$) | $C_5 HG_{32}$ triol (pg L$^{-1}$) | $C_5 HG_{30}$ triol (ng g$^{-1}$) | $C_5 HG_{32}$ triol (ng g$^{-1}$) | TOC (%) |
| 1 | 15.02 | -30.56 | 29/08/14 | 5500 | 36.4 | 18.0 | 0.00 | 1.7 ± 0.4 | 0.2 ± 0.1 | 0.6 ± 0.0 |
| 2 † | 14.35 | -32.58 | 30/08/14 | 6300 | 36.5 | ns | ns | † | † | † |
| 3 | 13.16 | -36.21 | 31/08/14 | 5190 | 36.4 | 24.6 | 0.00 | 3.3 ± 0.8 | 0.4 ± 0.1 | 0.6 ± 0.0 |
| 4 † | 12.41 | -38.50 | 01/09/14 | 4810 | 36.2 | 40.9 | 0.00 | † | † | † |
| 5 | 10.83 | -40.47 | 02/09/14 | 4620 | 36.0 | 8.9 | 0.00 | 2.3 ± 0.9 | 0.3 ± 0.1 | 0.5 ± 0.1 |
| 6 † | 9.41 | -42.10 | 03/09/14 | 3610 | 36.1 | 27.3 | 0.00 | † | † | † |
| 7 | 7.52 | -44.28 | 04/09/14 | 4650 | 33.5 | 773 | 66.2 | 14.6 ± 6.8 | 1.7 ± 1.0 | 0.7 ± 0.0 |
| 8 | 6.49 | -45.45 | 05/09/14 | 4250 | 31.9 | 4837 | 469 | 9.7 ± 1.7 | 1.0 ± 0.1 | 0.6 ± 0.0 |
| 9 * | 5.60 | -46.40 | 06/09/14 | 3770 | 32.2 | 24.6 | 0.00 | 4.8 ± 0.6 | 0.4 ± 0.0 | 0.5 ± 0.0 |
| 10 * | 6.68 | -47.49 | 07/09/14 | 4080 | 31.3 | 13.3 | 0.00 | 6.8 ± 4.4 | 0.7 ± 0.2 | 0.7 ± 0.0 |
| 11 | 5.53 | -51.50 | 10/09/14 | 80 | 29.2 | 0.00 | 0.00 | 0.2 ± 0.1 | 0.01 ± 0.01 | 0.1 ± 0.0 |
| 12 | 6.07 | -52.46 | 10/09/14 | 70 | 35.4 | 3.01 | 0.00 | 0.3 ± 0.1 | 0.01 ± 0.01 | 0.3 ± 0.1 |
| 13 | 7.60 | -53.02 | 11/09/14 | 1000 | 32.8 | 31.1 | 0.00 | 7.4 ± 3.0 | 0.9 ± 0.4 | 1.2 ± 0.0 |
| 14 * | 9.53 | -51.32 | 12/09/14 | 4840 | 31.4 | 316 | 6.2 | 13.5 ± 1.4 | 1.3 ± 0.2 | 0.9 ± 0.0 |
| 15 † | 8.95 | -49.98 | 13/09/14 | 4660 | 32.7 | 565 | 24.4 | † | † | † |
| 16 | 10.22 | -51.88 | 14/09/14 | 4940 | 33.9 | 391 | 27.3 | 13.0 ± 6.1 | 1.6 ± 0.5 | 1.0 ± 0.1 |
| 17 | 9.90 | -53.27 | 15/09/14 | 4750 | 31.6 | 379 | 15.5 | 9.4 ± 3.0 | 0.9 ± 0.3 | 0.9 ± 0.1 |
| 18 † | 9.37 | -55.20 | 16/09/14 | 3590 | 33.2 | 611 | 67.2 | † | † | † |
| 19 † | 10.52 | -55.48 | 16/09/14 | 4180 | 32.8 | 390 | 34.5 | † | † | † |
| 20a | 11.27 | -54.16 | 17/09/14 | 4790 | 33.9 | 2.3 | 0.0 | 17.6 ± 7.0 | 1.4 ± 1.2 | 0.8 ± 0.0 |
| 20b † | 11.47 | -54.21 | 17/09/14 | 4830 | 34.2 | 67.7 | 0.0 | † | † | † |
| 21a | 13.02 | -54.67 | 18/09/14 | 5040 | 33.8 | 196 | 9.8 | 12.9 ± 1.7 | 1.6 ± 0.1 | 0.6 ± 0.0 |
| 21b † | 13.20 | -54.72 | 18/09/14 | 5170 | 34.8 | 249 | 6.5 | † | † | † |
| 22 | 14.80 | -55.18 | 19/09/14 | 5500 | 35.6 | 48.6 | 0.4 | 13.4 ± 4.5 | 2.4 ± 1.1 | 0.7 ± 0.1 |
| 23 | 15.79 | -57.05 | 20/09/14 | 5320 | 34.0 | 106 | 5.9 | 9.8 ± 3.5 | 1.4 ± 0.5 | 0.6 ± 0.0 |




Table 2. The additional SPM samples collected for high resolution depth profile at Sta. 10 (0.3 μm GF/F). * =
Deep chlorophyll maximum.

| Sampling depth (m) | Salinity | Temperature (ºC) | $C_5$ $HG_{30}$ triol (pg $L^{-1}$) | $C_5$ $HG_{32}$ triol (pg $L^{-1}$) | Sum (pg $L^{-1}$) |
|---|---|---|---|---|---|
| 20 | 35.3 | 28.6 | 5.6 | 0.0 | 5.6 |
| 50* | 36.4 | 27.3 | 9.6 | 0.0 | 9.6 |
| 200 | 35.2 | 11.4 | 108 | 0.3 | 108 |
| 400 | 34.7 | 7.4 | 24.4 | 0.1 | 24.5 |
| 600 | 34.6 | 6.3 | 29.0 | 0.1 | 29.1 |
| 800 | 34.6 | 5.1 | 22.5 | 0.4 | 22.9 |
| 1000 | 34.7 | 4.7 | 11.9 | 0.2 | 12.2 |
| 1200 | 34.8 | 4.8 | 12.6 | 0.2 | 12.8 |
| 1500 | 35.0 | 4.5 | 16.2 | 0.3 | 16.5 |
| 2000 | 35.0 | 3.4 | 18.3 | 0.3 | 18.6 |
| 2500 | 34.9 | 2.8 | 20.7 | 0.4 | 21.0 |
| 3000 | 34.9 | 2.4 | 22.3 | 0.3 | 22.6 |

**Supplement**

**Supplementary Figures**

**Figure S1.** Aquarius sea-surface salinity (SSS) satellite data (7 day composites), centered on (DD/MM/YY) a)

27/08/14, b) 03/09/14, c) 10/09/14, d) 17/09/14 and e) 24/09/14 showing highly dynamic plume location.

Approximate location of R/V *Pelagia* indicated with purple circle.

**Supplementary Tables**

**Table S1.** Phytoplankton composition from Chemtax software based on pigment analysis. Numbers represent

fraction of total Chl a. Fractions greater than 0.5 are highlighted in red and fractions between 0.1 and 0.2 are

highlighted in purple.

**Table S2.** Diazotroph enumeration data. 5 categories: three are symbionts (syms), with the diatoms *Rhizosolenia*

*clevei*, *Hemiaulus hauckii,* and *Guinardia cylindrus*, and two non-symbionts, *Trichodesmium* colonies and free

*Trichodesmium* trichomes. Units are trichomes $L^{-1}$.

**Table S3.** Glycolipid concentration data. Concentration of $C_5$ $HG_{30}$ triol and $C_5$ $HG_{32}$ triol (pg $L^{-1}$) along with

concentration of Chl a (ng $L^{-1}$) as measured by HPLC.