# Peer review of "C5 glycolipids of heterocystous cyanobacteria track symbiont abundance in the diatom *Hemiaulus hauckii* across the tropical north Atlantic"

_Biogeosciences, 2017_

## Referee Comment (RC1) · Anonymous Referee #1 · 26 Aug 2017

Two things seems to be lacking in this manuscript for full consideration as acceptable one: 1. Care in the Figure, I did not manage to find Figure 1 in the normal manuscript and in the Supplemental text 2. Quantitation of compounds seems to be confusing. Authors declare the use of internal standard and "correction factor" used relate to hexose glycolipid, nor value (or effort) seems to be declared for the actual compounds

---

## Author Comment (AC1) · 28 Aug 2017

Thank you for your initial pre-review comments. Indeed the lack of figures does not make for an auspicious start to the reviewing process. We are unsure what happened to the figures and contacted the journal, who suggested we uploaded the full document with figure below, as we have now done.

Regarding the quantification method we have applied in this study. As described in the method section, our samples contained a short-chain glycolipid standard, n-dodecyl-

$\beta$-D glucopyranoside, which is indeed a glycolipid with a hexose head group rather than a pentose one. In the majority of the studies of heterocyst-forming cyanobacteria that we have carried out, hexose glycolipids were the target compound. Hence N-dodecyl-b-D-glucopyranoside was developed for use as an internal standard as it is a commercially available analogue of the naturally occurring hexose glycolipids, yet contains a carbon chain that is not observed in nature. Furthermore we were previously able to isolate a hexose glycolipid, 1-(O-hexose)-3,25-hexacosanediol from cultures of free-living heterocyst-forming cyanobacteria (Bale, 2017) using preparative HPLC in order to determine a RRF between it and the internal standard. It is not currently possible to isolate enough of a naturally occurring pentose-glycolipid due to the limitations of culturing diatom-diazotroph associations. However, we do not expect massive differences in ionization efficiency between a hexose and a pentose glycolipid. Rather the lack of alcohols on the chain is a more likely factor driving any difference. The RRF of the internal standard and the hexacosanediol is therefore probably fairly realistic and, at this point, due to the lack of standards, our approach is the most realistic estimate of quantities there is right now. We will make it clear in any revised version of the manuscript that our quantification of the pentose-glycolipids is semi-absolute, as we used a similar, but not identical compound. But as the trends in the natural glycolipid concentration would not change with a different RRF, the conclusions in our manuscript regarding their suitability as biomarkers for diatom-diazotroph associations stand.

---

## Author Comment (AC2) · 28 Aug 2017

Please find here the figures and supplement for article bg-2017-300.

———————————————

[Figure]

[Figure]

HO — O — O — OH OH
HO                    21    OH

$C_6\ HG_{26}$ diol

HO — O — O — OH
HO                    21

$C_6\ HG_{26}$ keto-ol

HO — O — O — OH OH
HO                    25

$C_5\ HG_{30}$ diol

HO — O — O — OH OH OH
HO                    23

$C_5\ HG_{30}$ triol

HO — O — O — OH OH OH
HO                    25

$C_5\ HG_{32}$ triol

**Fig. 1.** Figure 1

[Figure]

**Fig. 2.** Figure 2

[Figure]

**Fig. 3.** Figure 3

[Figure]

**Fig. 4.** Figure 4

[Figure]

**Fig. 5.** Figure 5

[Figure]

**Fig. 6.** Figure 6

**Fig. 7.** Figure S1

| St-cast | | Bacillar | Chloroph | Chrysoph | Cryptoph | Cyanophy* _Synechococcus_ | Prochl | Dinoph | Prasinop | Prym | |
|---|---|---|---|---|---|---|---|---|---|---|---|
| 01-06 | 5 m GF/F | 0.00 | 0.00 | 0.00 | 0.00 | 0.28 | **0.51** | 0.03 | 0.00 | 0.18 | > 0.5 of total |
| 01-06 | BML GF/F | 0.00 | 0.00 | 0.00 | 0.00 | 0.36 | 0.48 | 0.03 | 0.00 | 0.14 | 0.1-0.2 of total |
| 01-06 | DCM GF/F | 0.00 | 0.01 | 0.41 | 0.03 | 0.00 | 0.42 | 0.02 | 0.05 | 0.07 | *Excluding Prochlorococcus |
| | | | | | | | | | | | |
| 02-02 | 5 m GF/F | 0.00 | 0.00 | 0.00 | 0.00 | 0.17 | **0.61** | 0.05 | 0.00 | 0.17 | |
| 02-02 | BML GF/F | 0.00 | 0.00 | 0.07 | 0.00 | 0.18 | **0.53** | 0.04 | 0.00 | 0.18 | |
| 02-02 | DCM GF/F | 0.00 | 0.01 | 0.36 | 0.04 | 0.11 | 0.31 | 0.04 | 0.04 | 0.09 | |
| | | | | | | | | | | | |
| 03-07 | 5 m GF/F | 0.00 | 0.00 | 0.08 | 0.00 | 0.18 | **0.55** | 0.04 | 0.00 | 0.16 | |
| 03-07 | BML GF/F | 0.00 | 0.00 | 0.00 | 0.00 | 0.22 | **0.56** | 0.02 | 0.00 | 0.20 | |
| 03-07 | DCM GF/F | 0.00 | 0.01 | 0.30 | 0.01 | 0.07 | **0.51** | 0.03 | 0.00 | 0.08 | |
| | | | | | | | | | | | |
| 04-01 | 5 m GF/F | 0.00 | 0.00 | 0.01 | 0.00 | 0.29 | 0.44 | 0.03 | 0.00 | 0.23 | |
| 04-01 | BML GF/F | 0.00 | 0.00 | 0.13 | 0.00 | 0.21 | 0.46 | 0.06 | 0.00 | 0.14 | |
| 04-01 | DCM GF/F | 0.00 | 0.00 | **0.51** | 0.03 | 0.01 | 0.35 | 0.02 | 0.08 | 0.00 | |
| | | | | | | | | | | | |
| 05-06 | 5 m GF/F | 0.00 | 0.00 | 0.00 | 0.00 | 0.29 | **0.50** | 0.04 | 0.00 | 0.16 | |
| 05-06 | BML GF/F | 0.00 | 0.00 | 0.00 | 0.00 | 0.46 | 0.36 | 0.05 | 0.00 | 0.13 | |
| 05-06 | DCM GF/F | 0.00 | 0.00 | 0.43 | 0.04 | 0.08 | 0.19 | 0.03 | 0.15 | 0.08 | |
| | | | | | | | | | | | |
| 06-01 | 5 m GF/F | 0.00 | 0.00 | 0.00 | 0.00 | 0.28 | 0.49 | 0.07 | 0.00 | 0.15 | |
| 06-01 | BML | 0.00 | 0.00 | 0.03 | 0.00 | 0.37 | 0.42 | 0.03 | 0.00 | 0.15 | |
| 06-01 | DCM GF/F | 0.00 | 0.00 | 0.44 | 0.03 | 0.08 | 0.18 | 0.00 | 0.13 | 0.14 | |
| | | | | | | | | | | | |
| 07-06 | 5 m GF/F | 0.00 | 0.00 | 0.06 | 0.00 | **0.50** | 0.19 | 0.03 | 0.00 | 0.22 | |
| 07-06 | BML GF/F | 0.00 | 0.01 | 0.00 | 0.00 | **0.81** | 0.00 | 0.01 | 0.01 | 0.16 | |
| 07-06 | DCM GF/F | 0.00 | 0.00 | 0.61 | 0.03 | 0.13 | 0.14 | 0.01 | 0.06 | 0.01 | |
| | | | | | | | | | | | |
| 08-06 | 5 m GF/F | 0.05 | 0.00 | 0.05 | 0.00 | 0.47 | 0.16 | 0.01 | 0.00 | 0.26 | |
| 08-06 | BML GF/F | 0.21 | 0.00 | 0.00 | 0.00 | **0.62** | 0.02 | 0.02 | 0.00 | 0.13 | |
| 08-06 | DCM GF/F | 0.00 | 0.00 | 0.15 | 0.02 | 0.01 | 0.47 | 0.02 | 0.06 | 0.27 | |

**Fig. 8.** Table S1

| Station | Longitude [degrees_east] | Latitude [degrees_north] | Depth (m) | yy-mm-dd | Rhizo syms | Hemiaulus syms | Trichodesmium colonies | Trichodesmium filaments | R. cylindrus syms |
|---|---|---|---|---|---|---|---|---|---|
| 1 | -30.56 | 15 | 4.7 | 28-Aug-14 | 6.0 | 21.4 | 0.9 | 13.7 | |
| 1 | -30.56 | 15 | 15.024 | 28-Aug-14 | 1.7 | 0.0 | 0.0 | 77.8 | |
| 1 | -30.56 | 15 | 27.637 | 28-Aug-14 | 1.7 | 0.0 | 4.3 | 32.5 | |
| 1 | -30.56 | 15 | 39.103 | 28-Aug-14 | 2.6 | 0.0 | 0.0 | 15.4 | |
| 1 | -30.56 | 15 | 81.919 | 28-Aug-14 | 0.0 | 0.0 | 3.4 | 104.3 | |
| 1 | -30.56 | 15 | 197.944 | 29-Aug-14 | 0.0 | 0.0 | 0.0 | 0.9 | |
| 2 | -32.58 | 14.35 | 18.042 | 28-Aug-14 | 1.7 | 10.3 | 0.0 | 4.3 | |
| 2 | -32.58 | 14.35 | 32.895 | 28-Aug-14 | 0.0 | 0.0 | 0.0 | 3.4 | |
| 2 | -32.58 | 14.35 | 33.438 | 28-Aug-14 | 2.6 | 3.4 | 0.0 | 0.0 | |
| 2 | -32.58 | 14.35 | 55.094 | 28-Aug-14 | 0.0 | 0.0 | 0.0 | 11.1 | |
| 2 | -32.58 | 14.35 | 123.245 | 28-Aug-14 | 0.0 | 0.0 | 0.0 | 1.7 | |
| 3 | -36.21 | 13.16 | 4.749 | 31-Aug-14 | 15.4 | 0.0 | 0.0 | 0.0 | |
| 3 | -36.21 | 13.16 | 44.183 | 31-Aug-14 | 3.4 | 17.1 | 0.9 | 5.1 | |
| 3 | -36.21 | 13.16 | 44.373 | 31-Aug-14 | 2.6 | 11.1 | 0.0 | 2.6 | |
| 3 | -36.21 | 13.16 | 57.806 | 31-Aug-14 | 1.7 | 0.0 | 0.9 | 0.9 | |
| 3 | -36.21 | 13.16 | 86.036 | 31-Aug-14 | 0.0 | 0.0 | 0.0 | 0.9 | |
| 4 | -38.5 | 13.41 | 14.549 | 1-Sep-14 | 2.6 | 0.0 | 1.7 | 0.0 | |
| 4 | -38.5 | 13.41 | 30.018 | 1-Sep-14 | 2.6 | 0.9 | 2.6 | 181.2 | |
| 4 | -38.5 | 13.41 | 44.581 | 1-Sep-14 | 0.0 | 0.0 | 0.9 | 41.0 | |
| 4 | -38.5 | 13.41 | 74.952 | 1-Sep-14 | 0.0 | 0.0 | 0.9 | 59.8 | |
| 4 | -38.5 | 13.41 | 96.802 | 1-Sep-14 | 1.7 | 0.0 | 0.9 | 6.8 | |
| 4 | -38.5 | 13.41 | 136.157 | 1-Sep-14 | 0.0 | 0.0 | 0.0 | 0.0 | |
| 5 | -40.47 | 10.83 | 4.299 | 2-Sep-14 | 0.0 | 0.0 | 3.4 | 23.9 | |
| 5 | -40.47 | 10.83 | 5.703 | 2-Sep-14 | 0.0 | 0.0 | 0.0 | 0.0 | |
| 5 | -40.47 | 10.83 | 26.701 | 2-Sep-14 | 0.0 | 7.7 | 2.6 | 88.9 | |
| 5 | -40.47 | 10.83 | 50.072 | 2-Sep-14 | 0.0 | 0.0 | 2.6 | 11.1 | |
| 5 | -40.47 | 10.83 | 76.843 | 2-Sep-14 | 0.0 | 0.0 | 1.7 | 58.1 | |
| 5 | -40.47 | 10.83 | 86.646 | 2-Sep-14 | 0.0 | 0.0 | 1.7 | 58.1 | |

**Fig. 9.** Table S2

| STATION | Lat | Long | m Depth | pg/L C5 HG30 triol | pg/L C5 HG32 triol | ng/L Chl-a by HPLC |
|---|---|---|---|---|---|---|
| 1 | 15.02 | -30.56 | 80 | 0 | 0 | 370.9 |
| 1 | 15.02 | -30.56 | 40 | 14.11 | 0 | 55.4 |
| 1 | 15.02 | -30.56 | 5 | 18.03 | 0 | 37.2 |
| 2 | 14.35 | -32.58 | 65 | 0 | 0 | 402.2 |
| 2 | 14.35 | -32.58 | 33 | 5.32 | 0 | 38.4 |
| 2 | 14.35 | -32.58 | 5 | na | na | 37.9 |
| 3 | 13.16 | -36.21 | 90 | 8.92 | 0 | 146.4 |
| 3 | 13.16 | -36.21 | 35 | 23.41 | 0 | 43.3 |
| 3 | 13.16 | -36.21 | 5 | 24.6 | 0 | 41.9 |
| 4 | 12.41 | -38.5 | 83 | 7.02 | 0 | 215.6 |
| 4 | 12.41 | -38.5 | 30 | 13.27 | 0 | 61 |
| 4 | 12.41 | -38.5 | 5 | 40.92 | 0 | 49 |
| 5 | 10.83 | -40.47 | 80 | 6.9 | 0 | 335.7 |
| 5 | 10.83 | -40.47 | 40 | 6.42 | 0 | 54.4 |
| 5 | 10.83 | -40.47 | 5 | 8.871 | 0 | 34.7 |
| 6 | 9.41 | -42.1 | 90 | 1.18 | 0 | 330.8 |
| 6 | 9.41 | -42.1 | 15 | 15.06 | 0 | 60.7 |
| 6 | 9.41 | -42.1 | 5 | 27.27 | 0 | 42.9 |
| 7 | 7.52 | -44.28 | 60 | 169.29 | 8.49 | 236.3 |
| 7 | 7.52 | -44.28 | 15 | 778.72 | 82.92 | 126.6 |
| 7 | 7.52 | -44.28 | 5 | 772.56 | 66.19 | 115.5 |
| 8 | 6.49 | -45.45 | 52 | 196.75 | 4.65 | 214.7 |
| 8 | 6.49 | -45.45 | 10 | 4753.05 | 505.66 | 470.2 |
| 8 | 6.49 | -45.45 | 5 | 4836.92 | 468.39 | 164.7 |
| 9 | 5.6 | -46.4 | 200 | 20.66 | 0 | na |
| 9 | 5.6 | -46.4 | 40 | 38.06 | 0 | 301.9 |
| 9 | 5.6 | -46.4 | 5 | 24.55 | 0 | 242.5 |
| 10 | 6.68 | -47.49 | 200 | 126.98 | 11.81 | na |
| 10 | 6.68 | -47.49 | 50 | 45.81 | 0 | 255.9 |
| 10 | 6.68 | -47.49 | 9 | 9.02 | 0 | 224.7 |
| 10 | 6.68 | -47.49 | 5 | 13.28 | 0 | 187 |
| 11 | 5.53 | -51.49 | 15 | 17.68 | 0 | 176.2 |
| 11 | 5.53 | -51.49 | 5 | 0 | 0 | 132.6 |
| 12 | 6.07 | -52.46 | 15 | 15.56 | 0 | 326 |
| 12 | 6.07 | -52.46 | 5 | 3.01 | 0 | 126.2 |
| 13 | 7.6 | -53.02 | 35 | 28.72 | 0 | 89.2 |
| 13 | 7.6 | -53.02 | 3 | 31.07 | 0 | 301 |

**Fig. 10.** Table S3

---

## Referee Comment (RC2) · Anonymous Referee #1 · 29 Aug 2017

many thanks for Figure 1 implementation. I found it in the material. A comment about it that need further modification before acceptance is that stereochemistry of sugars as indicated is not justified for the method adopted in the manuscript. So please eliminate these details from figures for pentose glycolipids. Comments by authors about quantitation of C5 glycolipids are convincing, however I did not find any mention in the actual manuscript. Some pieces of the answer to me SHOULD be added in suitable parts in the manuscript. I suggest M&M and Discussion parts be modified according to what authors declares in the answer to me. After this modification the manuscript could be

accepted for publication.

---

## Author Comment (AC3) · 1 Sep 2017

We thank the reviewer for their final feedback and for recommending our manuscript for publication. As suggested we will modify the stereochemistry of the sugars in Figure 1 and make the edits to the quantification description in the method section.

---

## Referee Comment (RC3) · Anonymous Referee #2 · 24 Sep 2017

Bale and colleagues provide a very valuable data set aimed at interrogating the biomarker potential of heterocyst glycolipids with a pentose headgroup (C5 HGs) for diatom-diazotroph associations (DDAs). To establish such a diagnostic relationship is relevant for the reconstruction of the occurrence of DDAs in the fossil record. Thus the topic of the manuscript is definitely of great interest, and both the samples selected and analysis performed are ideally suited.

My main concern with the manuscript is that description and discussion of results are sometimes not as clear as they could be:

1. In the result section I would recommend to group the different stations, as already done to a certain degree in paragraph 3.4. ("high-salinity open ocean sites", "coastal-shelf stations", etc.). The naming of individual stations with only their number, paired with a very detailed description of concentrations at each of them, makes it hard to filter out the most relevant trends.

2. Short chain (C26) C5 HGs were initially described by Wörmer et al. (2012) in freshwater systems and a culture, before the description of longer chain C5 HGs by Schouten et al. (2013) in symbionts. As Wörmer et al (2012) only described C26 HGS, I would generally recommend to clearly differentiate between long- and short-chain C5 HGs throughout the text, e.g. in the conclusions "long-chain C5 HGs provide a robust, reliable method for detecting DDAs". Such a differentiation would make the authors' statements much more robust, as it eliminates potential interference from the short chain C5 HGs.

3. More importantly, I think that the discussion of the correlation between long-chain C5 HGs and different DDAs and free-living cyanobacteria needs to be improved to solidify the claim of a diagnostic relationship. For example, cross-plots and regression curves should be shown, instead of only stating r and p values. Based on these values alone, actually a strong correlation of C5 HGs is also observed with Trichodesmium colonies, and this harms the proposed biomarker potential. Even though I may share the authors' opinion that this regression might be coincidental and due to the co-ocurrence of Trichodesmium and DDAs at certain stations, a better effort to demonstrate the specific correlation between DDAs and C5 HGs is mandatory. In this sense I would for example recommend to pay special attention to the values which deviate from the regression. Assuming that several source organisms for C5 HGs exist, the fact that one potential producer (e.g. Rhizoselenia symbionts) is not abundant when C5 HGs are highly concentrated does not imply that it is not a potential source organism, as other producers may be present (e.g. Hemiaulus symbionts). This concept is hinted at in l. 281-290, but should be expanded. On the other hand, abundance of an organism

without corresponding HG abundance (e.g. maxima of Trichodesmium colonies ) is a much more robust factor to rule out a potential source organism. In this sense it might also be interesting to plot a combined regression line for all DDAs vs C5 HGs.

4. Finally, the authors claim that the analyzed compounds are ribose-containing, but I couldn't find any description of how the sugar moiety has been characterized. If they haven't been described I would rather use the term pentose (as hexose is used for the C6 compounds).

Another issue is that the manuscript preparation sometimes seems a little careless, and I would appreciate a thorough revision. Some minor comments and edits include:

Text is sometimes indented, sometimes not.

l.12-13: "have a thickened cell walls" please correct use of singular/plural

l.14: use singular form "cyanobacterium" or plural verb form "make"

l.43: please specify that you are referring to heterocystous cyanobacteria "all hetero-cystous, non-symbiotic cyanobacteria"

l.45: It might be better to already mention here that short chain C5 HGs have been described in a non-symbiotic cyanobacterial culture, not only C6 HGs.

l.52: It is confusing to state that the "first study of the C5 HGs in the natural environment" was Bale et al. 2015 while providing the fact that "HGs with a C5 sugar moiety" were identified in freshwater environments three years earlier.

l.111: Station number is missing, maybe 10?

l.114: add "each", "For each sediment"

l.120: "freeze dried filtered seawater". I guess the lipids are extracted from the filters, not from the filtered seawater, right?

l.145: Just out of interest, have the authors tried to increase flow to shorten analysis

time? 0.2 ml/min seems slow for a UHPLC system

l.149: at which m/z is resolution measured?

l.178: is 36.3 a value for salinity?

l.180: "(Fig. 3c,d))" close parentheses

l.180: "NO3+NO2" (no subscript for "+")

l.212: may be rephrased: "Free Trichodesmium trichomes were broadly distributed (Fig. 4d) and often occurred…"

l.266: delete space: "Hemiaulus hauckii-Richelia"

l.290-293: The separation of DDAs depending on salinity with the current data is unclear, as the authors state. Therefore I would delete this topic and also the corresponding figure.

l.304-308: I think the sampling-volume explanation is a little confusing. Couldn't the authors just state that sensitivity of che chemical biomarker method is much more sensitive than the microscopic approach?

l.347: please rephrase to avoid the term "vegetal"

l.352: add "regarding" or similar: "difference regarding the limit of detection"

l.600: "Trichodesmium" should be in italics

l.603: use "dashed" instead of "broken"?

l.611-612, Table 1: "*" is not defined. Please define BMWL and DCM, even though they are already defined in the text, table should be informative on its own. Same for figure S1, where actually BML is used.

l.629: are Trichodesmium colonies expressed as colonies or trichomes/ml?

Figures: Please use larger fonts.
Figure 2-4: I think it would be better to place the axis legend (e.g. Salinity in figure 2) to the right, instead of on top of the figure. Especially in fig 3 and 4 this makes it easier to identify what is shown.

Figure 4: (d) is used twice, for panel (d) and what should be panel (e). Why is C32 C5 HG shown as %? Wouldn't it be more informative to show concentration?

---

## Author Comment (AC4) · 3 Oct 2017

We thank the reviewer for their thoughtful and constructive review. We are pleased he/she found our data set "valuable" and that "the topic of the manuscript is definitely of great interest". We found the reviewer's edits useful and have addressed their points below.

1. In the result section I would recommend to group the different stations, as already done to a certain degree in paragraph 3.4. ("high-salinity open ocean sites", "coastalshelf stations", etc.). The naming of individual stations with only their number, paired with a very detailed description of concentrations at each of them, makes it hard to filter out the most relevant trends.

We appreciate this idea of grouping the stations and will edit as per suggested.

2. Short chain (C26) C5 HGs were initially described by Wörmer et al. (2012) in freshwater systems and a culture, before the description of longer chain C5 HGs by Schouten et al. (2013) in symbionts. As Wörmer et al (2012) only described C26 HGS, I would generally recommend to clearly differentiate between long- and short-chain C5 HGs throughout the text, e.g. in the conclusions "long-chain C5 HGs provide a robust, reliable method for detecting DDAs". Such a differentiation would make the authors' statements much more robust, as it eliminates potential interference from the short chain C5 HGs.

Indeed this is an important differentiation, we will edit to make this clear throughout the manuscript that the C5 HGs associated with DDAs have C30 and C32 chains as opposed to the C26 chain seen in the study of Wörmer et al. (2012).

3. More importantly, I think that the discussion of the correlation between long-chain C5 HGs and different DDAs and free-living cyanobacteria needs to be improved to solidify the claim of a diagnostic relationship. For example, cross-plots and regression curves should be shown, instead of only stating r and p values. Based on these values alone, actually a strong correlation of C5 HGs is also observed with Trichodesmium colonies, and this harms the proposed biomarker potential. Even though I may share the authors' opinion that this regression might be coincidental and due to the co-ocurrence of Trichodesmium and DDAs at certain stations, a better effort to demonstrate the specific correlation between DDAs and C5 HGs is mandatory. In this sense I would for example recommend to pay special attention to the values which deviate from the regression. Assuming that several source organisms for C5 HGs exist, the fact that one potential producer (e.g. Rhizoselenia symbionts) is not abundant when C5 HGs are

highly concentrated does not imply that it is not a potential source organism, as other producers may be present (e.g. Hemiaulus symbionts). This concept is hinted at in l. 281-290, but should be expanded. On the other hand, abundance of an organism without corresponding HG abundance (e.g. maxima of Trichodesmium colonies ) is a much more robust factor to rule out a potential source organism. In this sense it might also be interesting to plot a combined regression line for all DDAs vs C5 HGs.

We agree that presenting the data as regression curves would strengthen our assertion that long-chain C5 HGs track the abundance of certain marine DDAs. We realize that only stating r and p values from Pearson Correlations doesn't give a full picture of the data. We have plotted the regressions as suggested (see attached Figure). We will include them in the supplement of the revised manuscript and describe them in the text. Indeed, looking at them brings an extra dimension to the results/discussion. While the number of Hemiaulus symbionts has the best correlation with the C5 HG concentration (r2 = 0.62), that of the Trichodesmium colonies isn't far off (r2 = 0.48). However, there is a clear outlier in the Hemiaulus symbiont vs. C5 HG concentration plot, which is station 8 at 10 m water depth. As the DDAs and Trichodesium are all surface dwellers (upper 5 m) we assume that this point related to dead/detrital HGs, whereas viable DDAs were not visible under the microscope. Hence we also plotted the four regressions for only surface data (n=19). Here the correlation between the number of Hemiaulus and the C5 HG concentration is stronger still (r2 = 0.97), and again so is that of the Trichodesmium colonies (r2 = 0.94). On closer examination of these regressions it is clear that one station, again station 8, with unusually high levels of both Hemiaulus symbionts and Trichodesmium colonies (station 8) drives both these correlations. Removal of station 8 from the regressions showed that the number of Hemiaulus symbionts has yet again the best correlation with the C5 HG concentration (r2 = 0.67) whereas the correlation with Trichodesmium colonies has disappeared (n=0.03), as would be expected. Interestingly in this sample subset there is a stronger correlation between Rhizosolenia symbionts and the C5 HG concentration (r2 = 0.56). We will include this information in the revised discussion.

4. Finally, the authors claim that the analyzed compounds are ribose-containing, but I couldn't find any description of how the sugar moiety has been characterized. If they haven't been described I would rather use the term pentose (as hexose is used for the C6 compounds).

We did not include this information in this manuscript as it was published previously but the sugar was identified as part of the study of Schouten et al. 2013. "Comparison of the retention time and mass spectrum of an authentic methyl ribose standard established that the C5 sugar was ribose, though the stereo-configuration of the sugar could not be determined. However, since L-ribose does not occur in nature and is only produced synthetically (e.g. Hu et al., 2011) we assume that the sugar moiety is D-ribose. Thus, glycolipid VII was identified as 1-(O-ribose)-3,27,29-triacontanetriol." We will include clearer reference to this structural information arising from the previous study in our manuscript.

Another issue is that the manuscript preparation sometimes seems a little careless, and I would appreciate a thorough revision. Some minor comments and edits include:

Text is sometimes indented, sometimes not.

This was our understanding of the BGS style. Don't indent first paragraphs but then indent thereafter. We will double check the indenting requirements and further check our indentations.

l.12-13: "have a thickened cell walls" please correct use of singular/plural

We will make this edit.

l.14: use singular form "cyanobacterium" or plural verb form "make"

We will make this edit.

l.43: please specify that you are referring to heterocystous cyanobacteria "all heterocystous, non-symbiotic cyanobacteria"

We will make this edit.

l.45: It might be better to already mention here that short chain C5 HGs have been described in a non-symbiotic cyanobacterial culture, not only C6 HGs.

We will mention this here.

l.52: It is confusing to state that the "first study of the C5 HGs in the natural environment" was Bale et al. 2015 while providing the fact that "HGs with a C5 sugar moiety" were identified in freshwater environments three years earlier.

We will edit this to make it clear that C5 HGs have been detected twice previously in a marine and fresh water environment, but that is the first study in the marine environment that correlates their presence with DDA cell counts.

l.111: Station number is missing, maybe 10?

Yes it should have been Station 10. We edit to correct this mistake.

l.114: add "each", "For each sediment"

We will make this edit.

l.120: "freeze dried filtered seawater". I guess the lipids are extracted from the filters, not from the filtered seawater, right?

This is correct. We will edit this phrase.

l.145: Just out of interest, have the authors tried to increase flow to shorten analysis

When developing the LC-MS method we examined the flow rate and found it to be suitable for separating a wide range of IPLs and maximizing the ionization stability. We do not discuss the LC-MS method development in this manuscript as it was not within the scope of the paper. We appreciate the reviewer's suggestion and will keep it in mind when reviewing the method for further application.

l.149: at which m/z is resolution measured?

The resolution stated in our method (70,000 rpm) is for m/z 200. We will include this information in our revised manuscript.

l.178: is 36.3 a value for salinity?

We will insert 'salinity of' into this sentence

l.180: "(Fig. 3c,d))" close parentheses

We will make this edit.

l.180: "NO3+NO2" (no subscript for "+")

We will make this edit.

l.212: may be rephrased: "Free Trichodesmium trichomes were broadly distributed (Fig. 4d) and often occurred

We will make this edit.

i.266: delete space: "Hemiaulus hauckii-Richelia"

We will make this edit.

l.290-293: The separation of DDAs depending on salinity with the current data is unclear, as the authors state. Therefore I would delete this topic and also the corresponding figure.

As the reviewer finds this topic and figure unclear we will remove the figure (6) and simplify this text to clarify our point.

l.304-308: I think the sampling-volume explanation is a little confusing. Couldn't the authors just state that sensitivity of the chemical biomarker method is much more sensitive than the microscopic approach?

We will edit this section to make it briefer and clearer to the reader

l.347: please rephrase to avoid the term "vegetal"

We will replace this word with "planktonic organisms".

l.352: add "regarding" or similar: "difference regarding the limit of detection"

We will make this edit.

l.600: "Trichodesmium" should be in italics

We will make this edit.

l.603: use "dashed" instead of "broken"?

We will make this edit.

l.611-612, Table 1: "*" is not defined. Please define BMWL and DCM, even though they are already defined in the text, table should be informative on its own. Same for figure S1, where actually BML is used.

We will make these edits to the Table and Fig S1.

l.629: are Trichodesmium colonies expressed as colonies or trichomes/ml?

We will make it clear in this Supplement legend that Trichodesium was enumerated both in terms of Trichodesium colonies L-1 and as free Trichodesium trichomes L-1.

Figures: Please use larger fonts.

We will increase font size in figures

Figure 2-4: I think it would be better to place the axis legend (e.g. Salinity in figure 2) to the right, instead of on top of the figure. Especially in fig 3 and 4 this makes it easier to identify what is shown.

We will make these edits to the figures

Figure 4: (d) is used twice, for panel (d) and what should be panel (e). Why is C32 C5 HG shown as %? Wouldn't it be more informative to show concentration?

We will edit the panel numbering and change the figure to show concentration of C32 C5 HG rather than percent of total HG.

[Figure]

A) All data (n=54)

Fig. 1. Regression curves for reply to Reviewer 2

---

## Referee Comment (RC4) · Anonymous Referee #2 · 9 Oct 2017

thank you very much for your reply and for thoroughly addressing my comments. I am looking forward to a revised manuscipt that surely will be of great interest to the community.

---

## Author Response (AR1)

Thank you for considering our manuscript "C$_5$ glycolipids of heterocystous cyanobacteria track symbiont abundance in the diatom Hemiaulus hauckii across the tropical north Atlantic" for publication in *Biogeosciences* as an article. We thank the two reviewers for their helpful and constructive edits and suggestions. We have made all requested edits in the manuscript as as well as some additional edits that were noted by us at this stage. Below we have replied to each of the reviewers' comments directly.

Yours sincerely,

Dr. Nicole Bale, on behalf of co-authors

**Review 1**

A comment about it that need further modification before acceptance is that stereochemistry of sugars as indicated is not justified for the method adopted in the manuscript. So please eliminate these details from figures for pentose glycolipids.

**We have included the following text at line 51: "a pentose sugar head group (C5), identified as D-ribose, rather than a C6 sugar (Fig. 1) (Schouten et al., 2013)."**

Comments by authors about quantitation of C5 glycolipids are convincing, however I did not find any mention in the actual manuscript. Some pieces of the answer to me SHOULD be added in suitable parts in the manuscript. I suggest M&M and Discussion parts be modified according to what authors declares in the answer to me.

**We have inserted the following text into the method section (lines 168-173): "It is not currently possible to isolate enough of a naturally occurring pentose-glycolipid due to the limitations of culturing sufficient diatom-diazotroph biomass. As we do not expect significant differences in ionization efficiency between a hexose and a pentose glycolipid we assume that the RRF of the internal standard and the hexacosanediol C$_6$ HG is similar to that of C$_5$ HGs. Nevertheless, quantification of the pentose-glycolipids should be interpreted with care".**

**Review 2**

1. In the result section I would recommend to group the different stations, as already done to a certain degree in paragraph 3.4. ("high-salinity open ocean sites", "coastal-shelf stations", etc.). The naming of individual stations with only their number, paired with a very detailed description of concentrations at each of them, makes it hard to filter out the most relevant trends.

**We appreciate this idea of grouping the stations and have introduced this as far as possible in the results section. However, the description of the data also needs to illustrate the cruise path in and out of the Amazon plume so some individual station descriptions have remained.**

2. Short chain (C26) C5 HGs were initially described by Wörmer et al. (2012) in freshwater systems and a culture, before the description of longer chain C5 HGs by Schouten et al. (2013) in symbionts. As Wörmer et al (2012) only described C26 HGS, I would generally recommend to clearly differentiate between long- and short-chain C5 HGs throughout the text, e.g. in the conclusions "long-chain C5 HGs provide a robust, reliable method for detecting DDAs". Such a differentiation would make the authors' statements much more robust, as it eliminates potential interference from the short chain C5 HGs.

**Indeed this is an important differentiation, we have edited throughout the manuscript to make it clear that the C5 HGs associated with DDAs have C30 and C32 chains as opposed to the C26 chain seen in the study of Wörmer et al. (2012).**

3. More importantly, I think that the discussion of the correlation between long-chain C5 HGs and different DDAs and free-living cyanobacteria needs to be improved to solidify the claim of a diagnostic relationship. For example, cross-plots and regression curves should be shown, instead of only stating r and p values. Based on these values alone, actually a strong correlation of C5 HGs is also observed with Trichodesmium colonies, and this harms the proposed biomarker potential. Even though I may share the authors' opinion that this regression might be coincidental and due to the co-ocurrence of Trichodesmium and DDAs at certain stations, a better effort to demonstrate the specific correlation between DDAs and C5 HGs is mandatory. In this sense I would for example recommend to pay special attention to the values which deviate from the regression. Assuming that several source organisms for C5 HGs exist, the fact that one potential producer (e.g. Rhizoselenia symbionts) is not abundant when C5 HGs are highly concentrated does not imply that it is not a potential source organism, as other producers may be present (e.g. Hemiaulus symbionts). This concept is hinted at in l. 281-290, but should be expanded. On the other hand, abundance of an organism without corresponding HG abundance (e.g. maxima of Trichodesmium colonies ) is a much more robust factor to rule out a potential source organism. In this sense it might also be interesting to plot a combined regression line for all DDAs vs C5 HGs.

**We will have included the regression curves as supplementary figure (Fig. S2) and discussed them in the revised discussion (e.g. lines 358-374).**

4. Finally, the authors claim that the analyzed compounds are ribose-containing, but I couldn't find any description of how the sugar moiety has been characterized. If they haven't been described I would rather use the term pentose (as hexose is used for the C6 compounds).

**We have included the following text at line 51: "a pentose sugar head group (C5), identified as D-ribose, rather than a C6 sugar (Fig. 1) (Schouten et al., 2013)."**

Text is sometimes indented, sometimes not.

**This was our understanding of the BGS style. Don't indent first paragraphs but then indent thereafter. We will double check the indenting requirements and further check our indentations.**

l.12-13: "have a thickened cell walls" please correct use of singular/plural

**We have made this edit.**

l.14: use singular form "cyanobacterium" or plural verb form "make"

**We have made this edit.**

l.43: please specify that you are referring to heterocystous cyanobacteria "all heterocystous, non-symbiotic cyanobacteria"

**We have made this edit.**

l.45: It might be better to already mention here that short chain C5 HGs have been described in a non-symbiotic cyanobacterial culture, not only C6 HGs.

**We have edited to make it clear that in almost all cases non-symbiotic cyanobacterial culture, produce mainly C6 HGs. We go on to mention the shorter C5 HGs in the next paragraph.**

l.52: It is confusing to state that the "first study of the C5 HGs in the natural environment" was Bale et al. 2015 while providing the fact that "HGs with a C5 sugar moiety" were identified in freshwater environments three years earlier.

**We have edited this to make it clear that we are referring to the first study of C30 and C32 diols and triols with C5 head groups in the natural environment**

l.111: Station number is missing, maybe 10?

**We have corrected this mistake.**

l.114: add "each", "For each sediment"

**We have made this edit.**

l.120: "freeze dried filtered seawater". I guess the lipids are extracted from the filters, not from the filtered seawater, right?

**This is correct. We have edited this phrase.**

l.145: Just out of interest, have the authors tried to increase flow to shorten analysis

**When developing the LC-MS method we examined the flow rate and found it to be suitable for separating a wide range of IPLs and maximizing the ionization stability. We do not discuss the LC-MS method development in this manuscript as it was not within the scope of the paper. We appreciate the reviewer's suggestion and will keep it in mind when reviewing the method for further application.**

l.149: at which m/z is resolution measured?

**We have included this information in our revised manuscript.**

l.178: is 36.3 a value for salinity?

**We have inserted 'salinity of' into this sentence**

l.180: "(Fig. 3c,d))" close parentheses

**We have made this edit.**

l.180: "NO3+NO2" (no subscript for "+")

**We have made this edit.**

l.212: may be rephrased: "Free Trichodesmium trichomes were broadly distributed (Fig. 4d) and often occurred

**We have made this edit.**

i.266: delete space: "Hemiaulus hauckii-Richelia"

**We have made this edit.**

l.290-293: The separation of DDAs depending on salinity with the current data is unclear, as the authors state. Therefore I would delete this topic and also the corresponding figure.

**We have removed this text and figure 6.**

l.304-308: I think the sampling-volume explanation is a little confusing. Couldn't the authors just state that sensitivity of the chemical biomarker method is much more sensitive than the microscopic approach?

**We have edited this section to make it briefer**

l.347: please rephrase to avoid the term "vegetal"

**We have replaced this word with "planktonic organisms".**

l.352: add "regarding" or similar: "difference regarding the limit of detection"

**We have made this edit.**

l.600: "Trichodesmium" should be in italics

**We have made this edit.**

l.603: use "dashed" instead of "broken"?

**We have made this edit.**

l.611-612, Table 1: "*" is not defined. Please define BMWL and DCM, even though they are already defined in the text, table should be informative on its own. Same for table S1, where actually BML is used.

**We have removed the * symbol (it originally denoted stations with additional SPM sampling at 200 m). We have included a definition of BMWL and DCM. We have also**

**included this definition in the legend for Table S1 and changed BML to BMWL in table itself.**

l.629: are Trichodesmium colonies expressed as colonies or trichomes/ml?

**We have edited Supplement Table 1 legend to show that Trichodesium was enumerated both in terms of Trichodesium colonies L$^{-1}$ and as free Trichodesium trichomes L$^{-1}$.**

Figures: Please use larger fonts.

**We have increased the font size in all figures.**

Figure 2-4: I think it would be better to place the axis legend (e.g. Salinity in figure 2) to the right, instead of on top of the figure. Especially in fig 3 and 4 this makes it easier to identify what is shown.

**We have made this edit to Figures 2-4.**

Figure 4: (d) is used twice, for panel (d) and what should be panel (e). Why is C32 C5 HG shown as %? Wouldn't it be more informative to show concentration?

**We have edited the panel numbering. We have chosen to keep C$_5$ HG$_{32}$ as percent of total HG rather than concentration in order to illustrate the pint we wished to discuss in the text.**

[revised manuscript text omitted]